psychology

face recognition, emotion recognition, masks, super-recognizers, face matching

**Author for correspondence:**
Eilidh Noyes
e-mail: e.noyes@hud.ac.uk

# The effect of face masks and sunglasses on identity and expression recognition with super-recognizers and typical observers

Eilidh Noyes[1], Josh P. Davis[2], Nikolay Petrov[2],
Katie L. H. Gray[3] and Kay L. Ritchie[4]

[1]Department of Psychology, University of Huddersfield, Huddersfield HD1 3DH, UK
[2]School of Human Sciences, University of Greenwich, London SE10 9LS, UK
[3]School of Psychology and Clinical Language Sciences, University of Reading, Reading RG6 6UR, UK
[4]School of Psychology, University of Lincoln, Lincoln, Lincolnshire, LN6 7TS, UK

EN, 0000-0001-8709-879X; JPD, 0000-0003-0017-7159;
KLHG, 0000-0002-6071-4588; KLR, 0000-0002-1348-760X

Face masks present a new challenge to face identification (here matching) and emotion recognition in Western cultures. Here, we present the results of three experiments that test the effect of masks, and also the effect of sunglasses (an occlusion that individuals tend to have more experienced with) on (i) familiar face matching, (ii) unfamiliar face matching and (iii) emotion categorization. Occlusion reduced accuracy in all three tasks, with most errors in the mask condition; however, there was little difference in performance for faces in masks compared with faces in sunglasses. Super-recognizers, people who are highly skilled at matching unconcealed faces, were impaired by occlusion, but at the group level, performed with higher accuracy than controls on all tasks. Results inform psychology theory with implications for everyday interactions, security and policing in a mask-wearing society.

## 1. Introduction

The human face provides us with a great deal of information about a person, and arguably the two most important pieces of information in a face are the person's identity and their emotional state [1]. Judgements of identity and emotion facilitate social interactions [2] and can inform identification procedures in policing and security contexts. In the light of the

current COVID-19 pandemic, some countries are instructing that people wear face masks [3]. Face masks occlude the lower features of a face, with the potential to impact on identity and emotion perception.

Most people can recognize the faces of their friends, family and favourite celebrities with ease despite variation in the appearance of these faces across different encounters [1,4–7]. Familiar face recognition is robust against image-level changes, meaning that the face can be recognized across changes in pose, lighting, illumination, camera distance and, in some cases, deliberate disguise [4,8–14]. It is often possible to identify familiar faces when parts of the face are occluded [15–17]. By contrast, identity comparisons for faces that we have not previously encountered, or have very little experience with, is error prone [8,10,12,13,18–20]. Any differences between the comparison images (e.g. differences in pose, illumination or expression) increases task difficulty (e.g. [21]) and these effects are aggravated by natural changes to appearance such as from ageing [22]. People frequently mistake two images of the same unfamiliar face as different people [4,23], and mistake different people of similar appearance for the same person [24]. The error-prone nature of unfamiliar face identification has important consequences in applied settings.

Face comparisons inform identifications in many policing and security scenarios. A typical task at border control, for example, involves comparing the person at the border with the image on their passport. This sort of unfamiliar face matching task has been replicated in various laboratory and live settings. A 10–20% error rate is typical in standard face matching tasks which present two images side by side on a computer screen and ask participants to decide whether the images show the same person or two different people [25]. Presentation of a live person and a face photograph does not yield improved performance [7,26,27], nor does experience with face matching in passport officers [28].

Past studies show that drastic changes to facial features such as changing a person's appearance to look deliberately unlike themselves, or like someone else, severely impairs unfamiliar face identification [13,16,17,29]. When faces are altered to include competing identity information, identification may also be severely impacted [30,31]. For example, when the top half of one face (target region) is aligned with the bottom half of another (distractor region), observers' perception of the target region is biased [32]. In sequential matching paradigms, this 'composite face effect' results in observers making errors when judging the target region, often reporting that the target regions are different when they are in fact the same [33]. The composite face effect is thought to be driven by holistic processing, where local features are processed as a unified whole [34].

A fully occluded face, such as a face occluded by a stocking mask, is extremely difficult to identify [35]. However, the effect of partial occlusion is less understood. The simple addition of props to a face (e.g. sunglasses or glasses) has also been found to reduce accuracy on unfamiliar face matching tasks [13,16,17,29,36,37], where props remove information about the identity by occluding features of the face. Previous studies which have attempted to assess the importance of specific facial features for face identification have typically grouped features generically as 'external features' (e.g. hair) or 'internal features' (e.g. eyes, nose, mouth [38–40]). The relative contribution of specific internal features of the face is unknown. For unfamiliar face matching, the occlusion of any facial feature will probably reduce performance, with task difficulty increasing as more of the face is covered [20]. The eye region is argued to be the most diagnostic cue for face identification [41,42]. Glasses partially occlude, and sunglasses fully occlude the eye region of the face. Two studies have shown that unfamiliar face matching is impaired when one image in the pair is unconcealed and the other wears glasses [36,37]. The performance was further reduced when the eye region in one image in the pair was fully occluded through sunglasses, thus suggesting a role for the eye region in unfamiliar face matching [36]. It has been suggested that the mouth region may also be useful for face identification. Mileva & Burton [43] found higher face matching accuracy for pairs of images presenting open-mouthed smiles compared with neutral facial expressions. The authors argue that a person's smile provides idiosyncratic information about their identity, and therefore helps with face matching. Thus, occluding the mouth region of the face ought to reduce face matching accuracy. A recent paper, examining surgical face masks only, superimposed masks on to existing images of celebrities and also on to images from a large face database [44]. They found reduced familiar and unfamiliar face matching when one or both images in the pair was masked, compared with both images being unconcealed [44]. To date, no studies have directly compared the effect of each of these conditions (occluding the eye region compared against occluding the mouth region) on face matching performance.

While unfamiliar face matching is error prone there are some people who perform well above typical levels—referred to as 'super-recognizers' (see [45] and [46] for reviews). At the group level, super-recognizers perform with consistently high accuracy [47–51]. There have been several attempts to explore potential underpinnings of superior face recognition ability, with some studies suggesting that

super-recognizers use different features of the face to inform their identification. One study [52] reported that super-recognizers fixate on the nose region of the face more often relative to controls during face matching tasks. Other studies have suggested that fixations to the eyes [53], or just below the eye region, is associated with optimal face identification [54]. To date, there has been no investigation of super-recognizers' performance on face matching tests in which parts of the face are occluded. However, in two memory-based experiments, Davis & Tamonyte [55] found that super-recognizers outperformed controls at the recognition of faces wearing balaclavas (eye region only visible), hats and sunglasses (eye region covered), as well as with no facial occlusion. In their first experiment, test phase 10-person line-ups were presented immediately after the first phase single face image 8 s familiarization trial. In Experiment 2, a single 1 min video was displayed in Phase 1. The delays between Phase 1 and viewing a video line-up in Phase 2 was at least one week. In both experiments, accuracy was highest in the no occlusion condition, and lowest in the balaclava condition; although effect sizes were larger for correct rejections of previously unseen faces, than when correctly identifying those viewed before. At an individual level, fixation patterns, and performance accuracy is not always consistent with that expected of a super-recognizer [48,50,51,56]. Any study of super-recognizer performance on concealed faces should consider the performance of super-recognizers at the group level, and also consider the spread of performance at the individual level.

In addition to providing us with identity information, faces provide a highly informative cue to individuals' emotions. Seven 'basic' expressions (anger, fear, disgust, neutral, happiness, sadness and surprise) are thought to be recognized universally [57–59]; however, see [60]. These emotional expressions are processed rapidly [61], and recognized accurately in neurotypical adults [62], irrespective of face familiarity [63]. Facial expressions can be decomposed into action units [64] where each expression is described as a configuration of muscle movements. Some emotional expressions are best described by action units that are mainly in the mouth region of a face, such as happiness, and others are mainly in the eye region, such as fear [64]. To investigate whether this information is used by observers to categorize emotion, researchers have presented partially occluded faces (using 'bubbles' or other shapes) and calculated which face regions are correlated with categorization accuracy for different expressions [65–67]. Generally, mouth regions are most informative for happy, surprised and disgusted expressions, whereas eye regions are most informative for fearful and angry expressions, and both regions are informative for sad and neutral expressions [65,66]. Not only are the regions that are informative for each expression more likely to be fixated in an emotional categorization task, occluding these informative regions also disproportionately impacts accuracy [68]. Removing half of the image (i.e. only presenting the top or bottom half of an emotional face) leads to similar findings, with happy and disgust being most recognizable from the bottom half of the face and anger, fear and sadness being most recognizable from the top half of the face [69]. However, limited research has addressed the effects of naturalistic occlusion on emotional expression judgements.

One exception investigated the impact of sunglasses and masks on emotion categorization [70]. In this study, sunglasses were added onto a validated set of emotional faces using image editing software. The face mask condition, however, consisted of a non-realistic grey ellipse being added to the mouth region of the faces. Although occluding some of the same region as a realistic face mask, the ellipse did not cover the nose in most of the images presented. The authors [70] found that adults classified each emotional expression (happy, sad, surprise, fear and anger) less accurately when sunglasses were added to the images than when the images were unaltered. Accuracy was reduced further by masks than by sunglasses when all emotional expressions were combined, but the reduction in accuracy for each expression was not reported for the mask condition.

It has been suggested that super-recognizers who outperform controls on face matching and memory tests, may also be superior at identifying emotional expressions. Rhodes *et al.* [71] report that a person's emotion and face recognition abilities correlate more strongly than emotion and car recognition ability. In addition, Connolly *et al.* [72] identified a positive relationship between scores on emotion and face recognition tests for all of the basic emotions other than happiness. Very recent work has linked face matching ability as scored by the Glasgow face matching task, a standardized test of face matching ability [25], with recognition ability for emotions anger, fear and happiness, and also for neutral faces [73]. No differences were observed for the recognition of disgust or sadness [73]. Links between face recognition ability and emotion recognition have also been observed at the other end of the face recognition ability spectrum—developmental prosopagnosia may also be associated with the impaired perception of emotional expressions [74]. Together these results suggest some degree of positive

relationship between emotion and identity recognition ability. It is unknown whether this will extend to concealed faces.

The effect of face concealment on familiar and unfamiliar face recognition, as well as emotion recognition, is increasingly relevant as many countries around the world recommend the wearing of face masks which cover the mouth and nose in an attempt to slow or stop the spread of COVID-19. Mask wearing raises new questions in face perception—how accurate will unfamiliar face matching be for masked faces? Will we still be able to recognize familiar faces and gauge how a person is feeling? Here, we provide the first comprehensive assessment of the effect of masks on familiar (Experiment 1) and unfamiliar face matching (Experiment 2), and expression categorization (Experiment 3). We compare face matching and expression recognition accuracy for unconcealed faces, faces wearing sunglasses and faces wearing mouth and nose covering face masks. We used sunglasses as a comparison for face masks for three reasons: (i) this allows for a comparison of concealment of the upper (sunglasses) and lower (face masks) parts of the face; (ii) there is some previous work using sunglasses on which we can base our predictions; and (iii) historically, sunglasses have been more commonly seen than face masks in Western countries. We also compare performance on all three tasks between control participants and super-recognizers. We predicted that both sunglasses and face masks would give rise to poorer face matching performance for unfamiliar faces. A recent paper found a detrimental effect of digitally added surgical masks on familiar face recognition [44], and so based on this we predicted that masks would give rise to poor familiar face matching accuracy. We also predicted that sunglasses and face masks would affect emotion categorization differently for different expressions. We predicted that expressions containing diagnostic information in the top half of the face, including fear and anger, would be more affected by sunglasses, whereas expressions containing diagnostic information in the bottom half of the face, including happiness and surprise, would be more affected by masks. Finally, we hypothesized that super-recognizers would outperform controls in all three tasks. In sum, here we provide the first study to directly compare face matching and emotion categorization performance for super-recognizers and typical observers for faces in no concealment, sunglasses and masks. Our study is novel in its use of real mask images (rather than computer-generated facial occlusions) in the matching tasks. Our matching task mask stimuli are representative of the types of masks worn during COVID-19.

## 2. Experiment 1—familiar face matching

Familiar face identification is robust against many forms of image manipulation [4,8–14]; however, drastic changes in the appearance of a familiar face can impair identification [13]. Super-recognizers are typically better at face identification tasks than controls [47,49,51,75,76]. The advantage extends to some types of concealed faces [55]. Experiment 1 tests the effect of masks and sunglasses on familiar face identification for control participants and super-recognizers.

Importantly, in all three studies presented here, we use two different groups of control participants. Our super-recognizers were recruited from a large database of participants used in previous research (e.g.[77–79]). These participants were originally attracted to take part in research after seeing media reports about a super-recognizer test. Consistent with previous studies [78–80], super-recognizers are defined as those scoring 40/40 on the Glasgow face matching test: short version (GFMT) [25] and 95+/102 (93%) on the Cambridge face memory test: extended (CFMT+) [81]. An estimated 2% of the population score 95 or above on the CFMT+ [81,82], while less than 5% achieve the maximum on the GFMT [25]. During that original database recruitment process, many participants did not meet the criteria to be classed as super-recognizers. These other participants have continued to take part in various face perception studies. Following research precedent [78,79,80], typical-ability participants invited from this second group who had previously scored within one standard deviation of the normal population mean on both the CFMT+ (i.e. 58–83: [49]) and GFMT (i.e. 28–36: [25]) were allocated to a 'practised controls' group. It is possible that practise on such tasks, and high levels of interest in participating in face recognition tasks, may themselves boost performance to a level closer to that of super-recognizers than unpractised controls [45]. In addition to the practised controls, we recruited one further control group of participants from the online recruitment platform, Prolific.co. These participants, although routinely completing surveys, do not routinely complete face processing tasks. This group had not been pre-screened using the GFMT or CFMT+, and so provide a random sample of the population [45].

| unconcealed | sunglasses | mask |
|---|---|---|
| 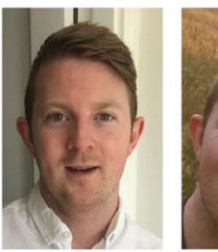 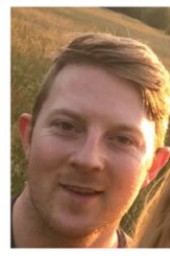 | 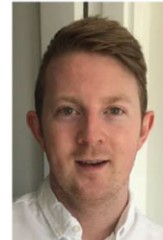 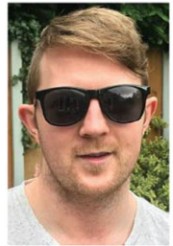 | 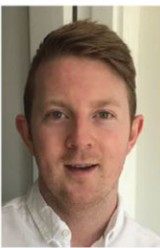 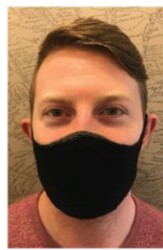 |

**Figure 1.** Example image pairs for each concealment condition. The above images all show the same person. (Copyright restrictions prevent publication of the images used in the experiment. Images in figure 1 are illustrative of the experimental stimuli and depict someone who did not appear in the experiments but has given permission for the images to be reproduced here).

## 2.1. Method

### 2.1.1. Participants

The unpractised control group (i.e. those who were randomly recruited and who may or may not have done face processing experiments previously) were recruited from Prolific.co. We included the specification that they must be resident in the UK so as to maximize the likelihood that they would recognize all of our celebrity faces, two of whom were likely only to be recognized in the UK. Each unpractised control participant was given £2.11 to compensate them for their time. The final sample was 102 (36 male, 65 female, one other; mean age 35 years; age range 18–63 years; 87.25% Caucasian).

The practised control group (i.e. those who have participated in the previous face processing experiments) were recruited from a large database of interested participants from the UK, and were not given monetary compensation. Members of this database are practised in that all have taken the CFMT+, the GFMT and at least one other face recognition test previously, and they may have been randomly selected to be invited to up to six online face processing projects per annum. Most projects provide debriefing feedback in terms of final test scores, so participants are normally roughly aware of their own ability, although no individual trial feedback is normally provided. In addition, no cross-referencing record is kept of whether they respond to invites or not. An initial sample of 306 participants took part. All claimed never to have taken the CFMT+ and GFMT prior to their database-stored score. Three were removed due to incomplete data. In order to match the unpractised control sample, we conducted the analyses on the first 102 participants to complete the tasks (30 male, 72 female; mean age 43 years; age range 21–72 years; 88.23% Caucasian). Practised controls had a mean GFMT score of 34.12/40 (s.d. = 1.89), and a mean CFMT+ score of 72.50 (s.d. = 7.09).

The super-recognizers were recruited from the same large database as the practised control participants, and will probably have been randomly invited to a similar number of face processing projects as the practised control group, and were not given monetary compensation. An initial sample of 159 participants took part, with one removed due to incomplete data. In order to match the two control samples, we conducted the analyses on the first 102 participants to complete the tasks (25 male, 77 female; mean age 39 years; age range 21–67 years; 91.17% Caucasian). Super-recognizers all scored 40/40 on the GFMT, and a mean CFMT+ score of 97.24 (s.d. = 1.75) as assessed in a previous battery of unpublished tests.

### 2.1.2. Stimuli

Four images of 12 celebrities were taken from the Internet in conditions (i) reference image (unconcealed), (ii) comparison image (unconcealed), (iii) sunglasses image, and (iv) mask image (figure 1). All images were gathered via Google Image search following the procedures used in previous research (e.g. [7,83]), with the only constraints being that the image should be good quality (i.e. not blurry), and show the face in a mostly front-facing view. There were no constraints in terms of facial expression displayed. All images were cropped to show head and shoulders at $380 \times 570$ pixels. The reference image was chosen as the more front-facing, neutral expression of the two unconcealed images, so as to approximate a passport-style image. A different identity 'foil' face image was selected for each identity to serve as the reference image in non-match trials. The foil identities were chosen to match the same verbal description as the target identity, e.g. 'young woman, blonde hair'. In all trials, the

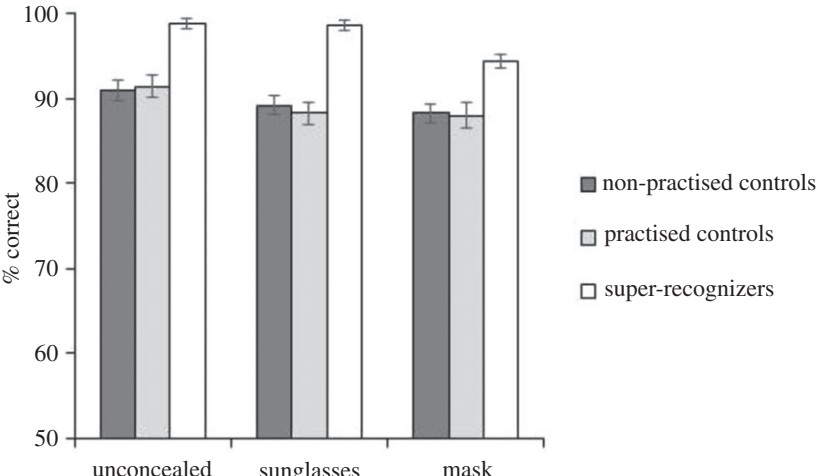

**Figure 2.** Data from Experiment 1, familiar face matching performance. Error bars show the within-subjects standard error [87].

image on the left was a reference image (either the reference image for the same ID trials of the celebrity, or the foil face for different identity trials). The image on the right was either the unconcealed comparison image, sunglasses image or mask image (figure 1).

### 2.1.3. Procedure

Participants completed the experiment online using the Qualtrics platform. Online tests of cognitive processing have become increasingly popular, and have been found to yield high-quality data that is indistinguishable from that collected in the laboratory [84–86]. In the familiar face matching task, participants were instructed that they would view pairs of face images and that their task was to decide whether the images were of the same person or two different people. Each face pair was presented side-by-side with the text 'Do you think these images depict the same or different people?' below the images. The response options presented below the text were 'Same' and 'Different'. The images and text remained on the screen until participants responded and clicked 'next' to see the next trial. On each trial, participants were also instructed to provide a confidence judgement for their same/different identity response for each trial—provided by the use of a sliding scale from 0 to 100. The confidence data are not presented in this paper. A practise trial with images not used in this experiment ensured that participants understood the paradigm. Identities were randomly assigned to conditions between participants, and each participant saw each identity only once, resulting in 12 trials. Participants saw two trials in each concealment condition (unconcealed, sunglasses, mask) for each trial type (match, non-match). The small number of trials in this experiment reflects both the difficulty of finding celebrities who would be familiar to most of our participants, and then finding images of those celebrities wearing face masks and sunglasses. At the end of the experiment, participants reviewed a list of names of celebrities and were asked to select all names for whom they would recognize the face. Participants received their accuracy score upon completion of all three experiments.

### 2.2. Results

We did not remove any trials based on participants' reported familiarity with the identities, and instead simply took familiarity as a group-level manipulation (as in [83] for example). Unpractised controls were familiar with a mean of 75.00% (9/12, s.d. = 2.30) celebrities, practised controls with 75.00% (9/12, s.d. = 3.52) celebrities, and super-recognizers with 91.67% (11/12, s.d. = 1.46) celebrities, therefore, we are satisfied that participants were, as groups, familiar with the celebrities presented in this experiment.

Mean accuracy for Experiment 1 is shown in figure 2. We began by analysing overall accuracy (collapsed across match and non-match conditions). A 3 (participant group: unpractised controls, practised controls, super-recognizers) × 3 (concealment: unconcealed, sunglasses, mask) mixed ANOVA showed a significant main effect of participant group $F_{2,303} = 20.89$, $p < 0.001$, $\eta_p^2 = 0.12$. Super-recognizers ($M = 97.22\%$) significantly outperformed both unpractised controls $M = 89.46\%$,

$t_{202} = 5.68$, $p < 0.001$, $d = 2.68$ and practised controls $M = 89.22\%$, $t_{202} = 8.01$, $p < 0.001$, $d = 2.87$, with no difference in performance accuracy between the unpractised and practised controls, $t_{202} = 0.15$, $p = 0.883$, $d = 0.07$. The ANOVA also showed a significant main effect of concealment $F_{2,606} = 5.12$, $p = 0.007$, $\eta_p^2 = 0.02$. Bonferroni corrected $t$-tests showed significantly poorer performance with pairs in which one face wore a mask ($M = 90.20\%$), compared with when both faces were unconcealed ($M = 93.71\%$), $t_{305} = 3.14$, $p = 0.006$, $d = 0.18$, and no difference between pairs in which one face was in sunglasses ($M = 91.99\%$) compared with when both were unconcealed, $p = 0.276$, and no difference in performance between the sunglasses and mask conditions, $p = 0.360$. The ANOVA showed a non-significant interaction $F_{4,606} = 0.67$, $p = 0.611$, $\eta_p^2 < 0.004$.

These results show that super-recognizers outperformed both of our control groups, with no difference between practised and unpractised controls. Signal detection analysis showed the same pattern of results (see electronic supplementary material, §S1). This result is consistent with prior research which found that super-recognizers outperformed control participants at identifying celebrities from poor quality images [75], and at matching celebrities with lookalikes in a pixellated face matching test [47]. In addition, both Davis *et al.* [75] and our present study found that super-recognizers claim to be familiar with more celebrities than control participants. This familiarity advantage may explain the difference in the performance of super-recognizers and controls.

The small overall decrease in performance for masked faces is driven predominantly by poorer performance on non-match trials (see electronic supplementary material, §S1) [88]. Our results show a subtle decrease in familiar face matching performance for masked faces compared with unconcealed faces. These results are broadly in line with a recent paper [44], although we found that masks gave rise to a smaller reduction in familiar face matching performance. When considering face identification in security settings, it is important to consider the effects of concealment on unfamiliar face recognition, therefore, we carried out a second experiment testing unfamiliar face matching with pairs of images showing unconcealed faces, as well as images wearing sunglasses or face masks.

# 3. Experiment 2—unfamiliar face matching

In this experiment, we investigated the effect of masks and sunglasses on unfamiliar face matching. Based on previous research, e.g. [36], we expected sunglasses to have a detrimental effect on face matching accuracy. Of particular interest was whether face masks would confer an additional detrimental effect on face matching.

## 3.1. Method

### 3.1.1. Participants, stimuli and procedure

The same participants who took part in Experiment 1 completed this experiment. Here, we used images of identities chosen to be unfamiliar to our participants. We collected images of 60 identities (30 female) which were publicly available on the Internet. Image selection and cropping were carried out in the same way as Experiment 1. As in Experiment 1, we collected two unconcealed images, one image wearing sunglasses and one image wearing a face mask which covered the mouth and nose. For each identity, we also collected one unconcealed image of a foil identity chosen to match the same verbal description as the target identity. The procedure was the same as that used in Experiment 1, but using 60 trials. Participants saw each identity once with the assignment of identities to conditions randomized across participants. Participants saw 10 trials in each concealment condition (unconcealed, sunglasses, mask) for each trial type (match, non-match).

## 3.2. Results

Unpractised controls reported recognizing a mean of 0 (s.d. = 0.24) identities, practised controls with 0 (s.d. = 0.24) identities and super-recognizers with 0 (s.d. = 0.29) identities, therefore, we are satisfied that the identities presented in this experiment were unfamiliar to our participants. Mean accuracy for Experiment 2 is shown in figure 3. We began by analysing overall accuracy. A $3 \times 3$ mixed ANOVA showed a significant main effect of participant group $F_{2,303} = 71.66$, $p < 0.001$, $\eta_p^2 = 0.32$, a significant main effect of concealment $F_{2,606} = 61.77$, $p < 0.001$, $\eta_p^2 = 0.17$ and a significant interaction $F_{4,606} = 3.19$, $p = 0.013$, $\eta_p^2 = 0.02$.

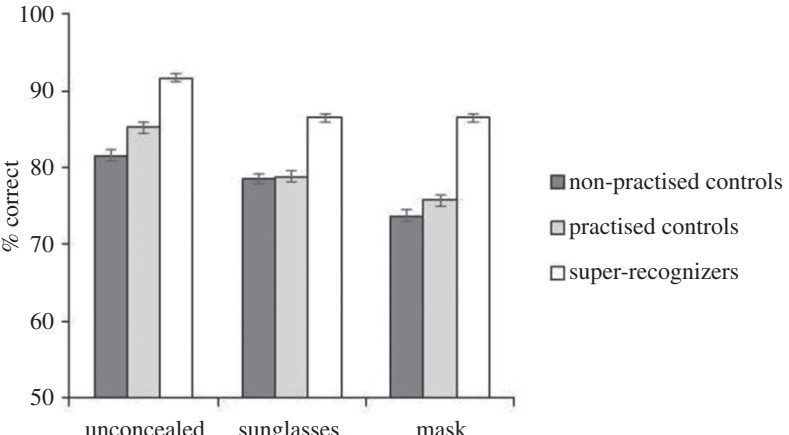

**Figure 3.** Data from Experiment 2, unfamiliar face matching performance. Error bars show the within-subjects standard error [74].

Bonferroni corrected *post hoc t*-tests revealed that super-recognizers performed significantly better than both control groups across all concealment conditions (all $ps < 0.05$), and that the practised control group outperformed the unpractised control group only for unconcealed faces ($p < 0.05$). The practised control group and super-recognizer groups performed with higher accuracy on unconcealed trials than on sunglasses trials (practised: unconcealed $M = 85.20$, sunglasses $M = 78.77$, $t_{101} = 5.05$, $p < 0.001$, $d = 0.50$; super-recognizers: unconcealed $M = 91.67$, sunglasses $M = 86.51$, $t_{101} = 5.08$, $p < 0.001$, $d = 0.57$). These same groups also performed with higher accuracy on unconcealed trials than mask trials (practised: mask $M = 75.74$, $t_{101} = 7.76$, $p < 0.001$, $d = 0.77$; super-recognizers: mask $M = 86.47$, $t_{101} = 5.71$, $p < 0.001$, $d = 0.57$). Unpractised controls showed no difference in their performance for unconcealed ($M = 81.52$) and sunglasses trials ($M = 78.53$), $p = 0.054$, but performed with higher accuracy for unconcealed trials than mask trials (mask $M = 73.73$), $t_{101} = 5.74$, $p < 0.001$, $d = 0.57$. Both control groups performed significantly better with sunglasses than masks (unpractised: $t_{101} = 3.62$, $p < 0.001$, $d = 0.36$, practised: $t_{101} = 2.33$, $p < 0.001$, $d = 0.23$). Super-recognizers showed no difference in performance for the sunglasses and mask conditions, corrected $p > 0.999$.

For unfamiliar face matching, super-recognizers outperformed both of our control groups, and performance was poorer with concealed faces compared with unconcealed faces. The reduction in performance for the control groups with masked faces was qualified by an increase in bias across all three participant groups for masked faces (see electronic supplementary material, §S2). For pairs of faces in which one image wore a mask, participants were more biased to respond 'non-match' or that the two images showed different people, as compared with unconcealed faces or sunglasses. These results are again broadly in line with a recent paper [37], but again our observed reduction in performance is smaller. It is important to note that performance across all groups and all conditions was consistently well above chance (50%), and so although concealment hindered performance, it did not completely destroy participants' ability to complete the task.

The super-recognizers in our study all scored 40/40 on the GFMT, and had a mean score of 97.2 on the CFMT+. Participants in the practised control group scored with lower accuracy than our super-recognizers on both of these standard tests. Some participants in the practised control group performed with very high accuracy on our tasks, and some participants in the super-recognizer group performed with accuracy levels well below the control mean. This result demonstrates noise in experimental testing [56] and speaks to the theoretical issues associated with the definition and selection of super-recognizers [45].

We had access to the GFMT scores and CFMT+ scores for both our practised control and super-recognizer group, and found significant correlations between performance on our tasks and the GFMT (figure 4), and our tasks and the CFMT+ (figure 5). This is consistent with the idea that performance on face tests tends to correlate with each other at the group level [89–91]. The scatterplots also clearly demonstrate spread in performance at the individual level for participants in each group. Our results reiterate the challenges associated with the definition of what it means to be a super-recognizer, and the importance of presenting super-recognizer data at both the group and individual level so that accurate conclusions on super-recognizer performance, and consistency of super-recognizer performance across tasks, can be drawn [45,50,51,56].

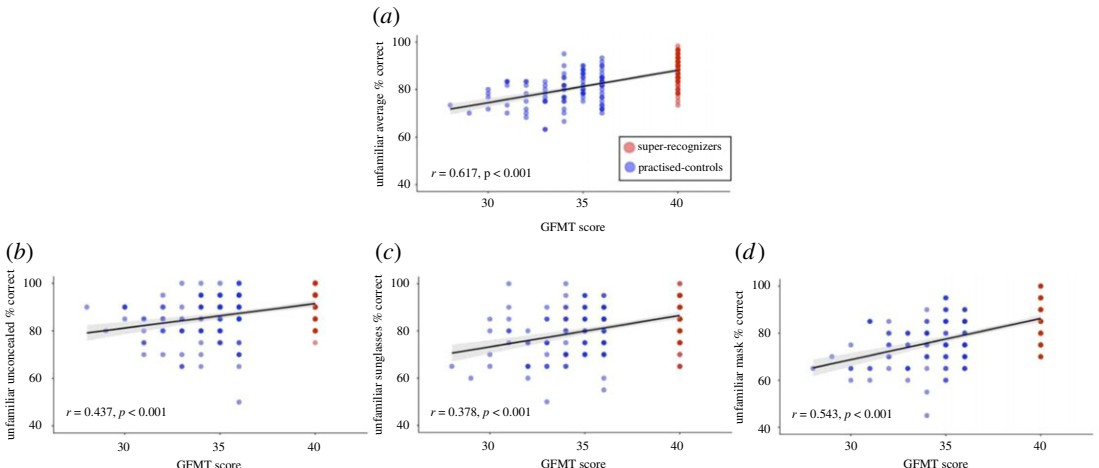

**Figure 4.** Scatterplots showing the relationship between GFMT performance and unfamiliar matching accuracy for (*a*) the average across all conditions, (*b*) unconcealed, (*c*) sunglasses and (*d*) mask conditions. Super-recognizers are given in red, and practised controls given in blue; individual data points are semi-transparent. 95% CI is given in grey.

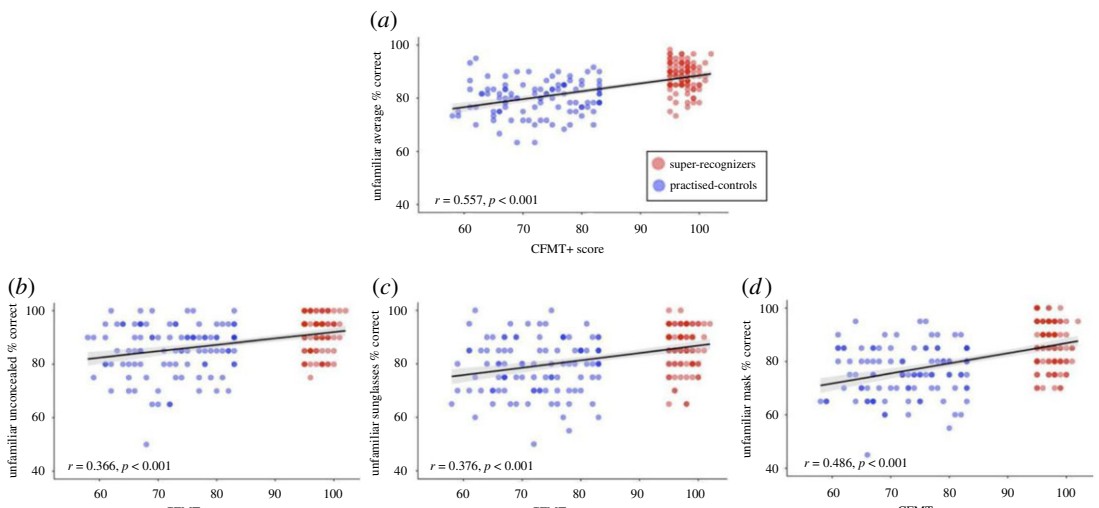

**Figure 5.** Scatterplots showing the relationship between CFMT+ performance and unfamiliar matching accuracy for (*a*) the average across all conditions, (*b*) unconcealed, (*c*) sunglasses and (*d*) mask conditions. Super-recognizers are given in red, and practised controls given in blue; individual data points are semi-transparent. 95% CI is given in grey.

In addition to posing a problem for identity recognition, face masks may also impair facial expression recognition. Our third experiment examined expression categorization with unconcealed faces, sunglasses and face masks.

# 4. Experiment 3—expression recognition

Typically, the whole face is used to categorize emotions [69]. However, different face regions are relatively more or less important for categorizing different emotional expressions; whereas some expressions contain critical diagnostic information in the eye region, others contain diagnostic information in the mouth region [64–67,69]. As sunglasses and masks block information from eye and mouth regions, respectively, the pattern of categorization impairment we find with these forms of concealment is likely to depend on the location of diagnostic information for each expression. Research on individual differences suggests that face identity recognition ability and expression recognition accuracy are related [71,72], indicating super-recognizers may have better emotion categorization than control participants.

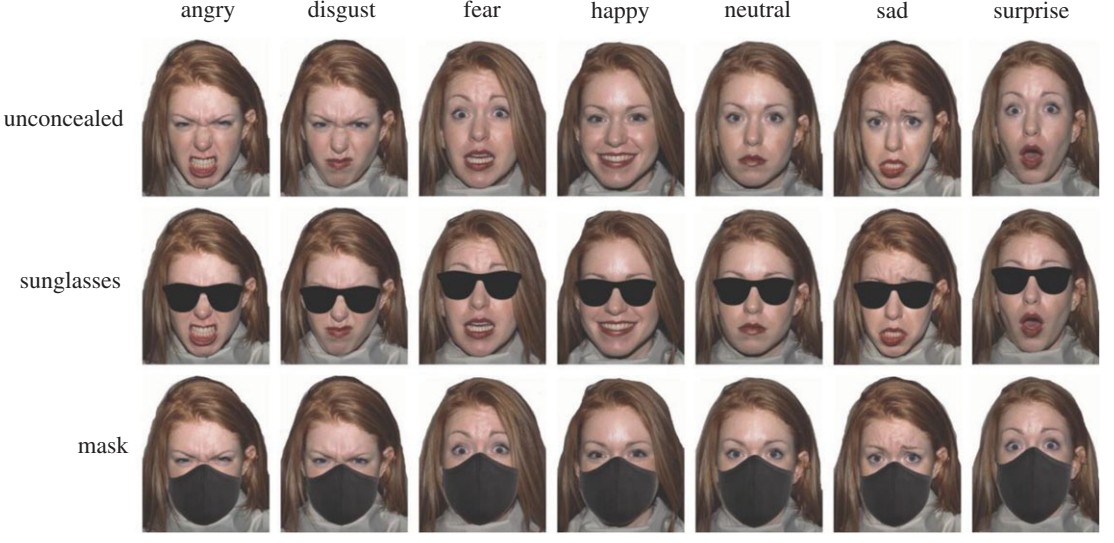

**Figure 6.** Example stimuli from Experiment 3 (emotional expression). An example from one identity is given across the different expressions and concealment conditions.

## 4.1. Method

### 4.1.1. Participants

Participants were taken from the same pool of participants as Experiments 1 and 2. Two participants from the unpractised control group (recruited via Prolific.co) did not have complete data and were removed, leaving 100 participants in this sample. To match the unpractised control group, the first 100 responders in the practised control group and super-recognizers were selected. One of the super-recognizers in this sample also did not have complete data, and so was replaced.

### 4.1.2. Stimuli

Images of 18 identities (nine female) displaying angry, disgust, fear, happy, neutral, sad and surprise expressions were selected from the NimStim face database [92] for their high emotional validity. Examples of sunglasses and face masks were sourced from the Internet, and added onto the images using Adobe Photoshop (figure 6). We chose to add sunglasses and masks to existing stimuli here (as opposed to using images of faces actually wearing sunglasses and masks as in Experiments 1 and 2) in order to ensure that the intensity and validity of the underlying expression was consistent across concealment conditions.

### 4.1.3. Procedure

At the beginning of the task, participants were instructed that they would see a face image and were asked to determine the emotion portrayed by the person in the image. Faces were presented as single images on the screen for 1000 ms, immediately followed by a choice of seven emotion response buttons. Participants chose when to proceed by pressing the next key. Seven practise trials, with unconcealed emotional expressions from identities not used in this experiment, ensured that participants understood the task. Each participant completed 63 trials (7 expressions × 3 concealment conditions × 3 repetitions). The three repetitions per condition consisted of different model identities. For each participant, and each emotion condition, nine identities were randomly selected from the possible 18 and were then randomly allocated to concealment condition. This enabled the restriction that the same identity could not be presented expressing the same emotion in different concealment conditions.

## 4.2. Results

The accuracy data for Experiment 3 is shown in figure 7 and table 1. We conducted a 3 (concealment: unconcealed, mask, sunglasses) × 7 (emotion: angry, fear, disgust, happy, neutral, sad, surprise) × 3

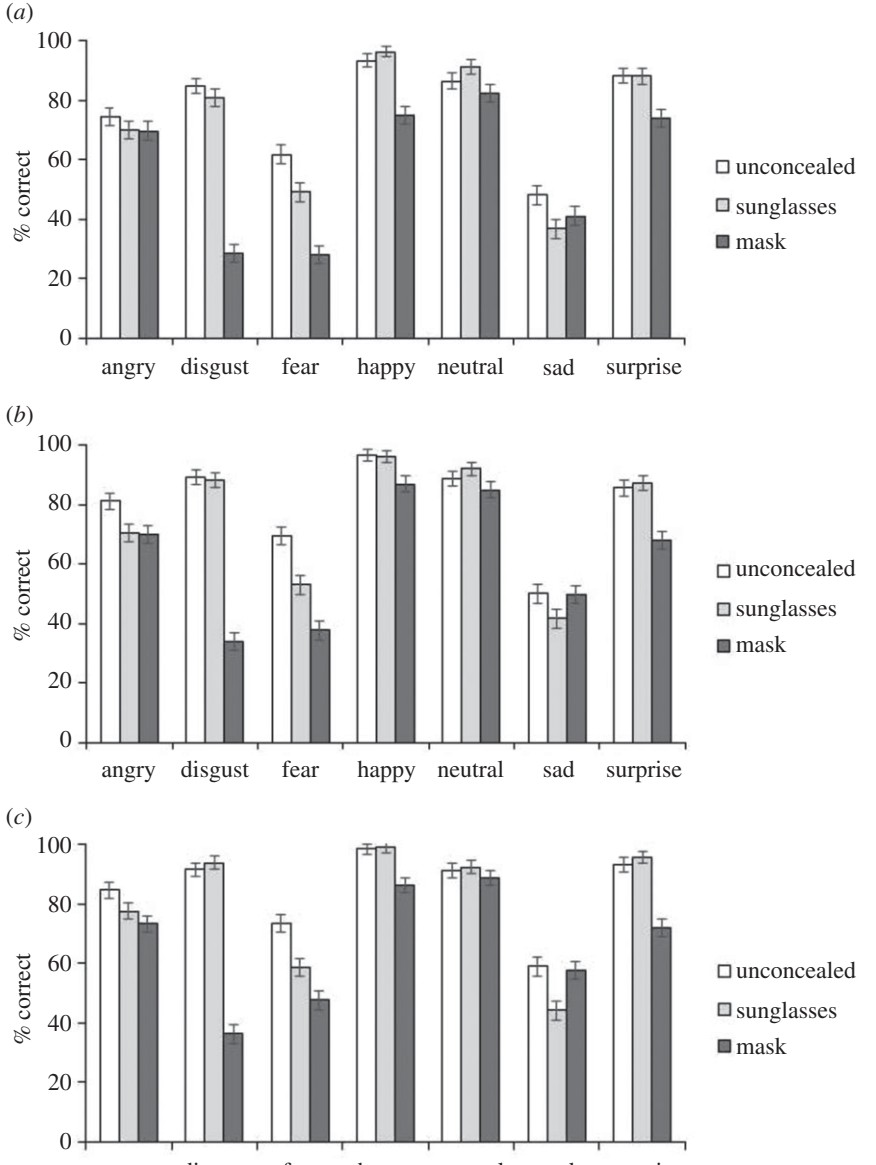

**Figure 7.** Data for Experiment 3, expression recognition. (*a*) Unpractised controls. (*b*) Practised controls. (*c*) Super-recognizers. Error bars show the within-subjects standard error [74].

(group: unpractised controls, practised controls, super-recognizers) mixed ANOVA. Where sphericity was violated, Greenhouse–Geisser adjustments are reported; *post hoc t*-tests are Bonferroni corrected. We found a significant main effect of group, $F_{2,297} = 26.09$, $p < 0.001$, $\eta_p^2 = 0.15$, whereby super-recognizers ($M = 76.95\%$) outperformed both the practised controls ($M = 72.46\%$) and unpractised controls ($M = 68.89\%$), and the practised controls also performed better than the unpractised controls (all $ps < 0.015$). There was also a significant main effect of emotion, $F_{5.1,1503.4} = 386.22$, $p < 0.001$, $\eta_p^2 = 0.57$, whereby all emotions differed from each other ($ps < 0.015$). Happy expressions were recognized best ($M = 92.07\%$), followed by neutral ($M = 88.63\%$), surprise ($M = 83.56\%$), angry ($M = 74.63\%$), disgust ($M = 69.74\%$), fear ($M = 53.19\%$) and then sad ($M = 47.56\%$) expressions. The ANOVA also showed a main effect of concealment $F_{1.9,577.5} = 381.66$, $p < 0.001$, $\eta_p^2 = 0.56$. Unconcealed expressions were recognized most accurately ($M = 80.48\%$), followed by emotions with sunglasses ($M = 76.32\%$), with expressions in the mask condition being most poorly recognized ($M = 61.51\%$; all $ps < 0.001$).

The pattern of errors made in emotional categorization tasks can be informative (e.g. [62,93]). These data are given in electronic supplementary material, §S3.

These main effects were subsumed within two significant interactions: an emotion by concealment interaction $F_{9.6,2856.2} = 74.98$, $p < 0.001$, $\eta_p^2 = 0.20$, and a small emotion by group interaction $F_{12,3636} = 2.24$, $p = 0.008$, $\eta_p^2 = 0.02$.

**Table 1.** Mean categorization performance (% correct) in Experiment 3. SRs, super-recognizers.

| | unconcealed % correct (s.d.) | | | masks % correct (s.d.) | | | sunglasses % correct (s.d.) | | |
|---|---|---|---|---|---|---|---|---|---|
| | SRs | practised controls | non-practised controls | SRs | practised controls | non-practised controls | SRs | practised controls | non-practised controls |
| angry | 84.67 (20.88) | 81.33 (23.36) | 74.33 (28.76) | 73.33 (21.19) | 70.00 (28.23) | 69.67 (29.24) | 77.67 (24.64) | 70.67 (26.50) | 70.00 (27.83) |
| disgust | 91.67 (15.98) | 89.33 (16.33) | 84.67 (20.88) | 36.33 (32.17) | 34.00 (29.12) | 28.67 (26.39) | 94.00 (13.72) | 88.33 (20.31) | 80.67 (26.03) |
| fear | 73.67 (26.08) | 69.33 (28.30) | 61.67 (31.91) | 47.67 (29.30) | 37.67 (32.36) | 28.00 (26.68) | 58.67 (28.87) | 53.00 (33.20) | 49.00 (29.38) |
| happy | 98.67 (6.56) | 96.67 (12.08) | 93.33 (18.35) | 86.33 (22.27) | 87.00 (21.13) | 75.00 (29.35) | 99.00 (5.71) | 96.33 (10.48) | 96.33 (14.13) |
| neutral | 91.33 (16.15) | 88.67 (21.30) | 86.33 (23.73) | 88.67 (19.08) | 85.00 (22.91) | 82.33 (26.15) | 92.33 (15.61) | 92.00 (17.16) | 91.00 (21.11) |
| sad | 59.00 (32.77) | 50.00 (31.25) | 48.00 (30.45) | 57.67 (28.37) | 49.67 (30.88) | 41.00 (32.77) | 44.33 (32.50) | 41.67 (33.63) | 36.67 (34.33) |
| surprise | 93.33 (15.71) | 85.67 (21.32) | 88.33 (19.17) | 72.00 (30.60) | 68.00 (26.77) | 73.67 (26.92) | 95.67 (13.12) | 87.33 (18.82) | 88.00 (19.26) |

In the emotion by concealment interaction, we ran one-way ANOVAs to probe the effect of concealment within each emotion. There was a significant effect of concealment in each of the expressions (angry: $F_{2,585.5} = 12.39$, $p < 0.001$, $\eta_p^2 = 0.04$; disgust: $F_{1.8,535.6} = 532.4$, $p < 0.001$, $\eta_p^2 = 0.67$; fear: $F_{1.9,576.5} = 92.46$, $p < 0.001$, $\eta_p^2 = 0.24$; happy: $F_{1.3,387.8} = 79.87$, $p < 0.001$, $\eta_p^2 = 0.21$; neutral: $F_{2,598} = 9.17$, $p < 0.001$, $\eta_p^2 = 0.03$; sad: $F_{2,598} = 12.06$, $p < 0.001$, $\eta_p^2 = 0.04$ and surprise: $F_{1.9,553.1} = 75.47$, $p < 0.001$, $\eta_p^2 = 0.20$).

For most expressions, performance was reduced on mask trials compared with unconcealed trials (angry: $t_{299} = 4.71$, $p < 0.001$, $d = 0.27$; disgust: $t_{299} = 28.98$, $p < 0.001$, $d = 1.67$; fear: $t_{299} = 14.22$, $p < 0.001$, $d = 0.90$; happy: $t_{299} = 8.95$, $p < 0.001$, $d = 0.52$ and surprise: $t_{299} = 9.84$, $p < 0.001$, $d = 0.57$). This was not the case for neutral ($p = 0.069$) or sad ($p = 0.696$) expressions, where there was no significant difference between unconcealed and mask conditions. Sunglasses significantly affected performance compared with the unconcealed condition, for some of the expressions (angry: $t_{299} = 4.03$, $p < 0.001$, $d = 0.23$; fearful: $t_{299} = 6.93$, $p < 0.001$, $d = 0.40$; sad: $t_{299} = 4.84$, $p < 0.001$, $d = 0.28$); sunglasses had no significant effect on performance for the other expressions ($ps > 0.108$). Expressive faces with sunglasses tended to be more accurately categorized than those with masks (disgust: $t_{299} = 26.83$, $p < 0.001$, $d = 1.55$; fear: $t_{299} = 6.45$, $p < 0.001$, $d = 0.37$; happy: $t_{299} = 9.74$, $p < 0.001$, $d = 0.56$; neutral: $t_{299} = 4.09$, $p < 0.001$, $d = 0.24$; surprise: $t_{299} = 10.06$, $p < 0.001$, $d = 0.58$), with two exceptions. The mask and sunglasses conditions did not significantly differ in the angry expressions ($p = 0.389$), and sad expressions were categorized more accurately in the mask than the sunglasses condition ($t_{299} = 3.43$, $p = 0.003$, $d = 0.20$).

To explore the emotion by group interaction, we ran one-way ANOVAs to probe the effect of group within each emotion. There was a significant effect of group in most of the expressions (angry: $F_{2,297} = 4.62$, $p = 0.011$, $\eta_p^2 = 0.03$; disgust: $F_{2,297} = 11.12$, $p < 0.001$, $\eta_p^2 = 0.07$; fear: $F_{2,297} = 12.56$, $p < 0.001$, $\eta_p^2 = 0.08$; happy: $F_{2,297} = 8.42$, $p < 0.001$, $\eta_p^2 = 0.05$; sad: $F_{2,297} = 8.07$, $p < 0.001$, $\eta_p^2 = 0.05$ and surprise: $F_{2,297} = 6.20$, $p = 0.002$, $\eta_p^2 = 0.04$), with no effect of group in the neutral expression ($p > 0.1$). We followed up these analyses with $t$-tests in all expressions except for neutral.

Super-recognizers outperformed non-practised controls in each of the expressions (angry: $t_{198} = 3.00$, $p = 0.012$; disgust: $t_{198} = 4.42$, $p < 0.001$; fear: $t_{198} = 5.42$, $p < 0.001$; happy: $t_{198} = 3.53$, $p = 0.002$; sad: $t_{198} = 4.12$, $p < 0.001$), except surprise ($p = 0.06$) and neutral. However, performance in the super-recognizer group was not significantly better than that of practised controls for most expressions (all $ps > 0.060$), except fear ($t_{198} = 2.41$, $p = 0.034$) and surprise ($t_{198} = 3.68$, $p < 0.001$), where super-recognizers did outperform practised controls. Practised controls tended to perform better than non-practised controls for some expressions (disgust: $t_{198} = 2.85$, $p = 0.01$; fear: $t_{198} = 2.43$, $p = 0.032$; happy: $t_{198} = 2.73$, $p = 0.014$); whereas for angry, sad and surprise expressions the two control groups did not statistically differ ($ps > 0.081$).

The pattern of errors made in emotional categorization tasks can be informative (e.g. [62,93]). These data are given in electronic supplementary material, §S3.

## 4.3. Associations across tasks

To explore whether performance on one task was associated with performance on the others, we ran between-task correlations (figure 8). We found a moderate correlation between performance on familiar and unfamiliar face matching ($r = 0.400$, $p < 0.001$), a small correlation between familiar face matching and emotion categorization ($r = 0.200$, $p = 0.001$) and a moderate correlation between unfamiliar face matching and emotion categorization ($r = 0.350$, $p < 0.001$). Please see the electronic supplementary material, §S4 for correlations broken down by condition, with and without the inclusion of the super-recognizer group.

## 4.4. Discussion

In several countries, the general public have been advised or instructed to wear face masks that cover the nose and mouth regions of the face to help reduce the transmission of COVID-19. Here, we tested the effect of masks on familiar face matching, unfamiliar face matching and emotion categorization as performed by humans. The matching task image pairs always consisted of one unconcealed face image, which was paired with an unconcealed, sunglasses or mask image. Matching accuracy was lower for the mask condition than for two unconcealed faces, regardless of face familiarity. On average performance reduced by 3.5% for masks compared with performance for unconcealed images when the faces were familiar, and by 7.5% for masks compared with unconcealed images when the faces were unfamiliar. There was no difference in accuracy for the mask and sunglasses conditions for

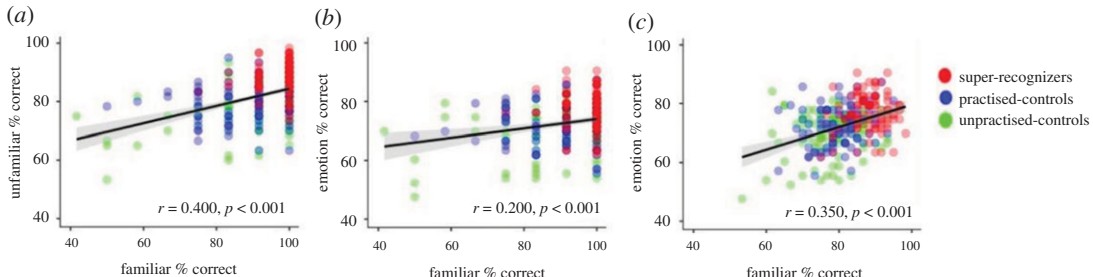

**Figure 8.** Correlations between average % correct scores on the familiar and unfamiliar face matching tasks (*a*), the familiar matching and emotion categorization tasks (*b*) and the unfamiliar matching and emotion categorization tasks (*c*). Individual data points are semi-transparent; 95% CI is given in grey. There is visible spread in performance at the individual level on all tasks (e.g. [50,94]).

familiar faces; however, for unfamiliar faces, this difference was 2.6% and the effect size was small. Our study has shown a smaller reduction in performance due to masks than was found in a recent paper [44]. This could be an artefact of the stimuli. Carragher & Hancock [44] superimposed surgical masks onto existing images, whereas we used real images of people wearing masks (and sunglasses), perhaps maintaining more of the structure of the face and aiding matching.

A different pattern of results was observed for familiar and unfamiliar faces. While both the sunglasses and mask conditions reduced matching performance for unfamiliar faces, occlusion of the eyes (sunglasses condition) did not impair performance on the familiar face matching task, whereas occlusion of the nose and mouth (mask condition) did reduce accuracy. Despite fewer trials in the familiar compared with unfamiliar experiment, this is unlikely to have driven differences between experiments. We present two possible explanations for the different pattern of results for familiar and unfamiliar faces. First, masks cover more features of a face than sunglasses. The eye region has previously been found to be the most diagnostic cue for face identification [41,42]; therefore, it is possible that the larger area concealed by masks leads to this relative drop in performance. Second, our participants might be more familiar with viewing sunglasses on a face than viewing masked faces in general, and may even have experience with viewing some of our famous faces in sunglasses. It will be interesting to track changes in face identification as the Western world becomes more familiar with viewing faces in masks.

The relatively small reduction in matching accuracy caused by masks or sunglasses is somewhat surprising given larger reductions in performance that have been observed for other forms of image manipulation. For example, Noyes & Jenkins [13] found that matching accuracy dropped by 35% when participants were presented with unfamiliar faces disguised to look unlike themselves, compared with performance for the same faces when they were presented without disguise. We argue that occlusion of facial features is less disruptive to identification than alteration of facial features, such as through make-up to create a disguise [13], contrast negation of facial features [95] or composite images [30,31]. Face images which have been altered contain information which can actively derail an identification by providing erroneous information about the appearance of features. For example, in the case of familiar face matching, an altered face may not match with the expected appearance for the identity [13]. When a face is unfamiliar, altered features may disrupt image-level comparisons and cause confusion for identifications. People may be better equipped to deal with occlusion, in which information from a face is removed rather than altered. This is perhaps achieved by 'filling in the gaps' created by occlusion with stored knowledge about the true appearance of features for a known identity, or by making use of the parts of the face which are visible.

The pattern of results was more complex in the emotion categorization task. In line with previous research that has linked different facial features as diagnostic for different emotions, the effect of masks and sunglasses varied across emotions. The emotional expressions that tend to have diagnostic information in the mouth region, such as disgust, happy and surprise [65,66], were most affected by the masks, showing a large reduction in categorization accuracy for disgust expressions in particular. The angry and fear expressions which tend to have diagnostic information in the eye region [65,66], were found to be affected by both sunglasses and masks; masks especially had a relatively large effect on fear categorization. When only the top half of a disgust face, or only the bottom half of a fear face is presented, there is a similar drop in performance [69]. The sad and neutral expressions, which have

diagnostic information across both the eyes and mouth regions [65,66] were differentially affected. Sad expressions were only disrupted in the sunglasses condition, suggesting the eye region is more critical for the accurate identification of this expression. The neutral expression accuracy actually improved in the sunglasses condition, suggesting that the information in the eye region is occasionally over-interpreted as emotional when neutral faces are unconcealed [96]. Although super-recognizers generally outperformed the unpractised control participants, and their mean scores were highest across most expressions, our analyses showed that their performance did not significantly differ from the practised controls' performance, with two exceptions. The super-recognizers had higher accuracy for fear and surprise expressions only.

Our study compared the performance of two groups of control participants and super-recognizers on each task. At the group level, super-recognizers outperformed controls on all tasks. This finding is consistent with previous work that shows that at a group level, super-recognizers consistently outperform controls on a range of face identification tasks [47–49,51,55,73,76]. However, also consistent with past work [48,49,51,55,75,76], there was a large spread in super-recognizers' performance at the individual level. All participants in our super-recognizer participant group had previously scored with 40/40 on the GFMT and with 97% accuracy on the CFMT+. High scores on these standard face recognition ability tests did not guarantee superior performance on our matching tasks. These results highlight the complexities associated with the definition of super-recognizers, and that caution must be exerted when interpreting group-level results, as these results will not necessarily transfer to the level of the individual. In Experiment 1, super-recognizers were slightly more likely to endorse being familiar with a larger number of identities than either control group. It is possible that this had an impact on the results in Experiment 1; however, this cannot be a factor in either of the other experiments. We did not record reaction times for Experiments 1–3, and so cannot comment on whether super-recognizers took longer to respond than control participants, benefiting from a speed–accuracy trade-off. Previous research has not found a difference in reaction times between super-recognizers and controls on face memory tasks [97], but comparing reaction times for these groups on a face matching task could be an interesting avenue for future research. The patterns of results from our two control groups were broadly similar. As we did not have a measure of how many face processing tasks each individual had previously participated in, we cannot say whether practise has or has not influenced the results of these experiments.

Here, we have considered face perception as performed by humans. Interestingly, a recent paper found that an automatic face recognition system, which uses a deep neural network to make similarity comparisons for face images, matched faces with lower accuracy when a face was presented in a mask than no mask, but only for some image types [44]. Specifically, ambient images of the face (similar to the images used in this experiment) with a superimposed mask, led to more errors than more controlled images with a superimposed mask [44]. It is unclear how the same algorithm would perform on a matching task that involved our ambient images for genuine mask wearing, and sunglasses wearing faces. This area requires more systematic investigation.

When we encounter a face in our everyday interactions, the face is typically accompanied by information from the person's body, voice and gait, which can all help inform our perceptions of the person's identity and current emotional state [98–100]. The results of our study are based on the presentation of still images. The additional cues that are available in live viewing environments will probably help overcome the effects of occlusion on identification and emotion perception. Masks do, however, remain an issue for unfamiliar face matching in situations where still images of the face are all that is available to inform an identification. While a new area of investigation, face identification and emotion recognition appear relatively robust against occlusion when considered against the effects of other forms of image manipulation which alter, rather than occlude facial features.

Ethics. This research was granted ethical approval from the University of Huddersfield School of Human and Health Sciences Ethics Committee (SREIC/2020/048).

Data accessibility. The datafiles for this paper are available from the online data depository site Dryad [101] (https://doi.org/10.5061/dryad.2280gb5pq).

Authors' contributions. E.N., J.P.D., N.P., K.L.H.G and K.L.R. designed the study. N.P. programmed the experiment. E.N., K.L.H.G. and K.L.R. analysed the data. E.N., J.P.D., N.P., K.L.H.G. and K.L.R. wrote the manuscript. All authors gave final approval for publication.

Competing interests. We declare we have no competing interests.

Funding. We received no funding for this study.

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
