## [Peer Review File · Royal Society Open Science]

Review History

RSOS-201169.R0 (Original submission)

Review form: Reviewer 1

Is the manuscript scientifically sound in its present form?

Yes

Are the interpretations and conclusions justified by the results?

Yes

Is the language acceptable?

Yes

Do you have any ethical concerns with this paper?

No

Have you any concerns about statistical analyses in this paper?

No

Recommendation?

Major revision is needed (please make suggestions in comments)

Comments to the Author(s)

Noyes et al. examined the extent to which recognition of famous face identity (E1), unfamiliar face identity (E2), and face emotion (E3) is influenced by disguising the eyes with sunglasses or the mouth with masks. This was compared in people with typical recognition ability and so-called super recognizers (SR). Recognition performance was measured in a face matching task (E1-2) and a forced-choice task, i.e., tasks with minimal memory demand. The paper is overall well written and easy to read. The figures are clear and the statistical approach for the critical analysis seem sound. Below I outline major and minor questions.

Major:

1. If I understand correctly, in Experiment 1 the SR were familiar with more of the famous faces than the control groups (i.e, SR: 11/12 versus controls 9/12 = 92% vs 75%). Could this also explain why the SR were better in the actual test (Figure 1)? That is, do they score higher because they are super recognizers or do they score higher because they happened to be more familiar with the people? If the latter, it would imply that super recognizers are only perhaps super recognizers for unknown faces (or in other words, everyone is super recognizers with familiar faces)?
2. For Experiments 1-3, what does the RT data look like? Could the increased accuracy for super recognizers be explained by also longer RTs, or are they both more accurate and faster?
3. For the interactions in Experiment 3, the authors choose to focus on the emotion x disguise rather than the emotion x group (or both). A rationale for why one is analyzed, but not the other, could be useful. Since the graph splits the data by group, it seems strange to skip any analysis where group is involved.
4. In the introduction, it could be useful to be more specific about the novel aspects of this study. For example, perhaps a brief summary sentence(s) in the final paragraph could state the novelty (and perhaps a rationale for why it is important)? The intro does mention novel aspects throughout, but some places it is a bit unclear what exactly is novel. For example, on the second page of the introduction (first paragraph), it is referred to research that occludes eyes with sunglasses [30], and another (preprint) study testing the effects of surgical masks [37], yet the concluding sentence of the paragraph reads "To date, no studies have compared the effect of occluding the eye region and occluding the mouth region on face matching performance". In this particular example, is novelty the face matching task (rather than some other recognition task) or is it that both conditions are included in the same study. In either case, a rationale for why this is important could also help the readers.
5. Practiced participants: When defining the 'practiced participants', the MS states " [Practiced participants] appeared to be particularly engaged and continued to take part in various face perception studies"? What does "particularly engaged" mean? How many experiments on average had the Ps already taken part in? Did they receive any face training during these experiments (or feedback for correct/incorrect trials)? If they only participated in standard experiments without feedback or training, then is there a rational reason to believe that they have become better at face recognition as a result of these studies (I can see that they are better at grasping the mechanics of a computerized face recognition task, but whether they have actually become better in face recognition is more difficult to apprehend)?

6. In the last sentence of stimuli section for Experiment 1, it states that Ps “saw two trials in each concealment condition (...) for each trial type”. Is it correct that there were only 4 trials per condition (2 match and 2 nonmatch), or was this per face identity (thus, 4*12 trials). If the latter, did face images repeat for each celebrity? If the former, why so few trials? Relatedly, it would be good to explicitly state whether each identity was shown in each condition.

7. For the correlation data (Figure 4, 5), why were not the non-practiced participants added?

8. In the summary for Experiment 2, it is stated that one could expect no overlap between SR and practiced controls. However, given that experiments are noisy (especially true for short experiments), I would in fact have expected that some super recognizers would overlap with practiced controls just due to chance, yet that they will be different at the group level. Also that said, if the argument is kept, then it could be beneficial to report the range of scores on the CFMT. Based on the argument, there should be no overlap between the highest practiced participant and lowest SR (unless I mistook something)?

Minor

9. page 4 (intro): “... task difficulty (e.g., [21] and ...”  bracket missing?

10. The introduction refers to findings with memory tasks, but a lack of similar findings with face matching tasks (e.g., page 2 of introduction, 2nd paragraph). It could be useful to be more specific about the memory task, if the goal is to distinguish the face matching task from the memory task. For example, what was the memory demand of the memory task (e.g., 1 second, 10 second, 30 minutes)? Note that even in a face matching task, there is a memory demand, since one can (presumably) only fixate/attend one face at the time.

11. In E1, perhaps parts of the stimuli section referring to trial structure would be more suited in the subsequent procedure section? That is, the trial structure is meaningless without knowing the paradigm, which is described in the following ‘procedure’ section.

12. Experiment 1 procedure. How long were the displays on for? Timed or until response? Would be good to be more specific about the procedure, and perhaps even add a figure (e.g., as panel A for Figure 1).

13. Experiment 1, results: When referring to number of celebrities known, it could be good to use percentage or at least show the maximum possible (e.g., $M = 75\%$ (9/12)). Also, if this is a mean score, it seems strange that there are no decimals?

Review form: Reviewer 2

Is the manuscript scientifically sound in its present form?

No

Are the interpretations and conclusions justified by the results?

Yes

Is the language acceptable?

Yes

Do you have any ethical concerns with this paper?

No

Have you any concerns about statistical analyses in this paper?

Yes

Recommendation?

Major revision is needed (please make suggestions in comments)

Comments to the Author(s)

In this manuscript the authors pursue the contemporary / obvious question of whether face occlusion impairs facial identity matching and emotion categorization, and whether such an impairment varies as a function of independently assessed face matching / recognition performance. Their results are as anticipated: less information available impairs performance (experiments 1 & 2), and more so if the information occluded is that which is diagnostic for the task (experiment 3). Below I detail main points, which I believe can be addressed.

Sincerely,

Meike Ramon

Analyses

The authors should fully exploit the fact that their observers performed all tasks, and include a more detailed, multivariate consideration of their performance patterns across all three experiments, under consideration of patterns at the individual level as they propose. Appropriate visualisations should also be provided in order for readers to fully appreciate the depth of available information at the single-subject level.

Correlational analyses should be provided throughout (not only selectively), as well as with/without inclusion of supers to address the question of whether they are driving linear relationships.

p.19., 2nd paragraph and Table 1: extremely cumbersome to work through; provide more straightforward approach and reporting.

Missing methodological information

There is no information provided regarding quality assurance measures implemented to control for viewing conditions across observers, which is particularly important given data acquisition was performed online. I would like to see the required information considering restrictions implemented to prevent participation from tablets and mobile phones, as well as details regarding implemented screen calibration procedures.

No detail is provided regarding the selection of image material for experiments 1 and 2 thereby preventing reproduction.

Describe how authors “ensure[d] that the intensity and validity of expressions was consistent across concealment conditions”.

Diagnostic approach at odds with authors own recent practice

In a recent manuscript (Dunn et al., 2020) the GFMT and CFMT+ were described as inappropriate for online assessment, motivating the creation of an(other) online test battery. Therefore, instead of framing the manuscript as a “Super-Recognizer” study the more appropriate approach is to consider only the measure which does not suffer from a ceiling effect (ie the CFMT+), and to investigate the relationship between face recognition performance and the manipulations implemented here.

Missing references

A number of relevant references, which should be cited are missing. These relate to the issue of familiar face processing including the aspect of partial information processing (cf. Ramon & Gobbini, 2018, *Vis Cogn*; Ramon, Busigny, Gosselin & Rossion, 2018, *Vis Cogn*), as well as those concerning the relationship across tests observed at the group vs. individual level (Stacchi et al.,

2020; Fysh et al., 2020), and notably the review of super-recogniser literature by Ramon et al., 2019 (a,b; BJP).

Minor points

- identification should not be used interchangeably with recognition and matching (eg first sentence in the abstract), and - in light of the reported manipulations implemented here - refer to the tasks with greater precision (e.g. p.4, l.31: comparisons of facial *identity*; given the nature of response measurement experiment 3 is actually an emotion categorisation, rather than recognition task).
- p.5, 1st paragraph: the related composite face effect should be discussed, as well as its implications that facial information in the bottom of the face may have an either facilitatory as well as inhibitory effect.
- remove p.5 2nd paragraph; eye tracking in normal controls and supers not appropriate, as information sampling does not imply usage, and the prior is not assessed here.
- range of performance scores should be provided for all samples reported (cf. p.8)
- remove reference to results reported purely in the Supplementary Material (cf. p.11, l.17f.)

Decision letter (RSOS-201169.R0)

Dear Dr Noyes

The Editors assigned to your paper RSOS-201169 "The effect of face masks and sunglasses on identity and expression recognition with super-recognisers and typical observers" have now received comments from reviewers and would like you to revise the paper in accordance with the reviewer comments and any comments from the Editors. Please note this decision does not guarantee eventual acceptance.

Please submit your revised manuscript and required files (see below) no later than 21 days from today's (ie 05-Nov-2020) date. Note: the ScholarOne system will 'lock' if submission of the revision is attempted 21 or more days after the deadline. If you do not think you will be able to meet this deadline please contact the editorial office immediately.

Please note article processing charges apply to papers accepted for publication in Royal Society Open Science (<https://royalsocietypublishing.org/rsos/charges>). Charges will also apply to papers transferred to the journal from other Royal Society Publishing journals, as well as papers submitted as part of our collaboration with the Royal Society of Chemistry

(<https://royalsocietypublishing.org/rsos/chemistry>). Fee waivers are available but must be requested when you submit your revision (<https://royalsocietypublishing.org/rsos/waivers>).

on behalf of Dr Bruno Rossion (Associate Editor) and Essi Viding (Subject Editor)
openscience@royalsociety.org

Reviewer comments to Author:
Reviewer: 1

Comments to the Author(s)

Noyes et al. examined the extent to which recognition of famous face identity (E1), unfamiliar face identity (E2), and face emotion (E3) is influenced by disguising the eyes with sunglasses or the mouth with masks. This was compared in people with typical recognition ability and so-called super recognizers (SR). Recognition performance was measured in a face matching task (E1-2) and a forced-choice task, i.e., tasks with minimal memory demand. The paper is overall well written and easy to read. The figures are clear and the statistical approach for the critical analysis seem sound. Below I outline major and minor questions.

Major:

1. If I understand correctly, in Experiment 1 the SR were familiar with more of the famous faces than the control groups (i.e, SR: 11/12 versus controls 9/12 = 92% vs 75%). Could this also explain why the SR were better in the actual test (Figure 1)? That is, do they score higher because they are super recognizers or do they score higher because they happened to be more familiar with the people? If the latter, it would imply that super recognizers are only perhaps super recognizers for unknown faces (or in other words, everyone is super recognizers with familiar faces)?
2. For Experiments 1-3, what does the RT data look like? Could the increased accuracy for super recognizers be explained by also longer RTs, or are they both more accurate and faster?
3. For the interactions in Experiment 3, the authors choose to focus on the emotion x disguise rather than the emotion x group (or both). A rationale for why one is analyzed, but not the other, could be useful. Since the graph splits the data by group, it seems strange to skip any analysis where group is involved.
4. In the introduction, it could be useful to be more specific about the novel aspects of this study. For example, perhaps a brief summary sentence(s) in the final paragraph could state the novelty (and perhaps a rationale for why it is important)? The intro does mention novel aspects throughout, but some places it is a bit unclear what exactly is novel. For example, on the second page of the introduction (first paragraph), it is referred to research that occludes eyes with sunglasses [30], and another (preprint) study testing the effects of surgical masks [37], yet the

concluding sentence of the paragraph reads “To date, no studies have compared the effect of occluding the eye region and occluding the mouth region on face matching performance”. In this particular example, is novelty the face matching task (rather than some other recognition task) or is it that both conditions are included in the same study. In either case, a rationale for why this is important could also help the readers.

5. Practiced participants: When defining the ‘practiced participants’, the MS states “ [Practiced participants] appeared to be particularly engaged and continued to take part in various face perception studies”? What does “particularly engaged” mean? How many experiments on average had the Ps already taken part in? Did they receive any face training during these experiments (or feedback for correct/incorrect trials)? If they only participated in standard experiments without feedback or training, then is there a rational reason to believe that they have become better at face recognition as a result of these studies (I can see that they are better at grasping the mechanics of a computerized face recognition task, but whether they have actually become better in face recognition is more difficult to apprehend)?

6. In the last sentence of stimuli section for Experiment 1, it states that Ps “saw two trials in each concealment condition (...) for each trial type”. Is it correct that there were only 4 trials per condition (2 match and 2 nonmatch), or was this per face identity (thus, 4*12 trials). If the latter, did face images repeat for each celebrity? If the former, why so few trials? Relatedly, it would be good to explicitly state whether each identity was shown in each condition.

7. For the correlation data (Figure 4, 5), why were not the non-practiced participants added?

8. In the summary for Experiment 2, it is stated that one could expect no overlap between SR and practiced controls. However, given that experiments are noisy (especially true for short experiments), I would in fact have expected that some super recognizers would overlap with practiced controls just due to chance, yet that they will be different at the group level. Also That said, if the argument is kept, then it could be beneficial to report the range of scores on the CFMT. Based on the argument, there should be no overlap between the highest practiced participant and lowest SR (unless I mistook something)?

Minor

9. page 4 (intro): “... task difficulty (e.g., [21] and ...”  bracket missing?

10. The introduction refers to findings with memory tasks, but a lack of similar findings with face matching tasks (e.g., page 2 of introduction, 2nd paragraph). It could be useful to be more specific about the memory task, if the goal is to distinguish the face matching task from the memory task. For example, what was the memory demand of the memory task (e.g., 1 second, 10 second, 30 minutes)? Note that even in a face matching task, there is a memory demand, since once can (presumably) only fixate/attend one face at the time.

11. In E1, perhaps parts of the stimuli section referring to trial structure would be more suited in the subsequent procedure section? That is, the trial structure is meaningless without knowing the paradigm, which is described in the following ‘procedure’ section.

12. Experiment 1 procedure. How long were the displays on for? Timed or until response? Would be good to be more specific about the procedure, and perhaps even add a figure (e.g., as panel A for Figure 1).

13. Experiment 1, results: When referring to number of celebrities known, it could be good to use percentage or at least show the maximum possible (e.g., $M = 75\%$ (9/12)). Also, if this is a mean score, it seems strange that there are no decimals?

Reviewer: 2

Comments to the Author(s)

In this manuscript the authors pursue the contemporary / obvious question of whether face occlusion impairs facial identity matching and emotion categorization, and whether such an impairment varies as a function of independently assessed face matching / recognition performance. Their results are as anticipated: less information available impairs performance (experiments 1 & 2), and more so if the information occluded is that which is diagnostic for the task (experiment 3). Below I detail main points, which I believe can be addressed.

Sincerely,

Meike Ramon

Analyses

The authors should fully exploit the fact that their observers performed all tasks, and include a more detailed, multivariate consideration of their performance patterns across all three experiments, under consideration of patterns at the individual level as they propose. Appropriate visualisations should also be provided in order for readers to fully appreciate the depth of available information at the single-subject level.

Correlational analyses should be provided throughout (not only selectively), as well as with/without inclusion of supers to address the question of whether they are driving linear relationships.

p.19., 2nd paragraph and Table 1: extremely cumbersome to work through; provide more straightforward approach and reporting.

Missing methodological information

There is no information provided regarding quality assurance measures implemented to control for viewing conditions across observers, which is particularly important given data acquisition was performed online. I would like to see the required information considering restrictions implemented to prevent participation from tablets and mobile phones, as well as details regarding implemented screen calibration procedures.

No detail is provided regarding the selection of image material for experiments 1 and 2 thereby preventing reproduction.

Describe how authors “ensure[d] that the intensity and validity of expressions was consistent across concealment conditions”.

Diagnostic approach at odds with authors own recent practice

In a recent manuscript (Dunn et al., 2020) the GFMT and CFMT+ were described as inappropriate for online assessment, motivating the creation of an(other) online test battery. Therefore, instead of framing the manuscript as a “Super-Recognizer” study the more appropriate approach is to consider only the measure which does not suffer from a ceiling effect (ie the CFMT+), and to investigate the relationship between face recognition performance and the manipulations implemented here.

Missing references

A number of relevant references, which should be cited are missing. These relate to the issue of familiar face processing including the aspect of partial information processing (cf. Ramon & Gobbini, 2018, *Vis Cogn*; Ramon, Busigny, Gosselin & Rossion, 2018, *Vis Cogn*), as well as those concerning the relationship across tests observed at the group vs. individual level (Stacchi et al., 2020; Fysh et al., 2020), and notably the review of super-recogniser literature by Ramon et al., 2019 (a,b; *BJP*).

Minor points

- identification should not be used interchangeably with recognition and matching (eg first sentence in the abstract), and - in light of the reported manipulations implemented here - refer to the tasks with greater precision (e.g. p.4, l.31: comparisons of facial *identity*; given the nature of response measurement experiment 3 is actually an emotion categorisation, rather than recognition task).
- p.5, 1st paragraph: the related composite face effect should be discussed, as well as its implications that facial information in the bottom of the face may have an either facilitatory as well as inhibitory effect.
- remove p.5 2nd paragraph; eye tracking in normal controls and supers not appropriate, as information sampling does not imply usage, and the prior is not assessed here.
- range of performance scores should be provided for all samples reported (cf. p.8)
- remove reference to results reported purely in the Supplementary Material (cf. p.11, l.17f.)

===PREPARING YOUR MANUSCRIPT===

===PREPARING YOUR REVISION IN SCHOLARONE===

Author's Response to Decision Letter for (RSOS-201169.R0)

See Appendix A.

RSOS-201169.R1 (Revision)

Review form: Reviewer 1

Is the manuscript scientifically sound in its present form?

Yes

Are the interpretations and conclusions justified by the results?

Yes

Is the language acceptable?

Yes

Do you have any ethical concerns with this paper?

No

Have you any concerns about statistical analyses in this paper?

No

Recommendation?

Major revision is needed (please make suggestions in comments)

Comments to the Author(s)

I thank the authors for their responses and revisions. Below are my remaining comments.

- Response to comments #1 and #2: I think these limitations should at least be explicitly mentioned in the general discussion. Essentially, the enhanced performance of super recognizers in E1 could be due to they simply being more familiar with the faces (i.e., not specific to super recognizer ability). Moreover, the enhanced performance of super recognizers in general could be due to them spending more time (e.g., being more meticulous) than the other participants on each image before making their judgement.
- Response to comment #5: Perhaps explicitly define practiced and non-practiced in the text. E.g., (if I understand the definition of each group correctly)
 - o practiced: participated in previous experiments from the lab
 - o non-practiced: randomly recruited Ps who may or may not have done face recognition studies in the past.

Moreover, it seems unclear whether the “practiced” participants have in fact completed more face-recognition studies than the “non-practiced” participants? If I understand correctly, the practiced have done more face recognition tests offered by this specific research group, but it seems possible that the non-practiced Ps could have completed face recognition tests from other groups? If there is no way to ascertain this, then I think this limitation should also be explicitly mentioned in the discussion.

- Response to comment #6: Given the few trials (which seems abnormal(?)), this might also be good to point out as a potential limitation in the discussion.

Decision letter (RSOS-201169.R1)

Dear Dr Noyes

The Editors assigned to your paper RSOS-201169.R1 "The effect of face masks and sunglasses on identity and expression recognition with super-recognisers and typical observers" have now received comments from reviewers and would like you to revise the paper in accordance with the reviewer comments and any comments from the Editors. Please note this decision does not guarantee eventual acceptance.

We do not generally allow multiple rounds of revision so we urge you to make every effort to fully address all of the comments at this stage, given that the Editors have opted to issue a further revision - indeed, if the Editors and reviewer are not satisfied with your revision it is unlikely a further revision will be offered. If deemed necessary by the Editors, your manuscript will be sent back to one or more of the original reviewers for assessment. If the original reviewers are not available, we may invite new reviewers.

Please submit your revised manuscript and required files (see below) no later than 21 days from today's (ie 05-Feb-2021) date. Note: the ScholarOne system will 'lock' if submission of the revision is attempted 21 or more days after the deadline. If you do not think you will be able to meet this deadline please contact the editorial office immediately.

on behalf of Dr Bruno Rossion (Associate Editor) and Essi Viding (Subject Editor)
openscience@royalsociety.org

Reviewer comments to Author:

Reviewer: 1

Comments to the Author(s)

I thank the authors for their responses and revisions. Below are my remaining comments.

- Response to comments #1 and #2: I think these limitations should at least be explicitly mentioned in the general discussion. Essentially, the enhanced performance of super recognizers in E1 could be due to they simply being more familiar with the faces (i.e., not specific to super recognizer ability). Moreover, the enhanced performance of super recognizers in general could be due to them spending more time (e.g., being more meticulous) than the other participants on each image before making their judgement.

- Response to comment #5: Perhaps explicitly define practiced and non-practiced in the text. E.g., (if I understand the definition of each group correctly)

o practiced: participated in previous experiments from the lab

o non-practiced: randomly recruited Ps who may or may not have done face recognition studies in the past.

Moreover, it seems unclear whether the "practiced" participants have in fact completed more face-recognition studies than the "non-practiced" participants? If I understand correctly, the practiced have done more face recognition tests offered by this specific research group, but it seems possible that the non-practiced Ps could have completed face recognition tests from other groups? If there is no way to ascertain this, then I think this limitation should also be explicitly mentioned in the discussion.

- Response to comment #6: Given the few trials (which seems abnormal(?)), this might also be good to point out as a potential limitation in the discussion.

===PREPARING YOUR MANUSCRIPT===

===PREPARING YOUR REVISION IN SCHOLARONE===

Author's Response to Decision Letter for (RSOS-201169.R1)

See Appendix B.

Decision letter (RSOS-201169.R2)

Dear Dr Noyes,

It is a pleasure to accept your manuscript entitled "The effect of face masks and sunglasses on identity and expression recognition with super-recognisers and typical observers" in its current form for publication in Royal Society Open Science. The comments of the reviewer(s) who reviewed your manuscript are included at the foot of this letter.

on behalf of Dr Bruno Rossion (Associate Editor) and Essi Viding (Subject Editor)
openscience@royalsociety.org

Appendix A

RSOS-201168

Title: The effect of face masks and sunglasses on identity and expression recognition with super-recognisers and typical observers.

Authors: Eilidh Noyes, Josh Davis, Nikolay Petrov, Katie L.H. Gray, Kay L. Ritchie.

26/11/20

Dear Editor,

Please find uploaded the revised version of the manuscript 'The effect of face masks and sunglasses on identity and expression recognition with super-recognisers and typical observers' with co-authors Josh Davis, Nikolay Petrov, Katie L.H. Gray and Kay L. Ritchie. We hope that you will consider this revised paper for publication in the journal Royal Society Open Science.

We would like to thank you and the reviewers for the comments on the manuscript, which we have addressed in turn.

Please do not hesitate to contact me if I may be of any assistance in processing the manuscript.

Yours Sincerely,

Eilidh Noyes

Eilidh Noyes, Ph.D.

Dept. of Psychology

University of Huddersfield

Huddersfield, UK

HD1 3DH

Email: e.noyes@hud.ac.uk

Tel: 01484472770

RSOS-201168

Title: The effect of face masks and sunglasses on identity and expression recognition with super-recognisers and typical observers

Authors: Eilidh Noyes, Josh Davis, Nikolay Petrov, Katie L.H. Gray, Kay L. Ritchie.

Reviewer comments to Author:

Reviewer #1:

Comments to the Author(s)

Noyes et al. examined the extent to which recognition of famous face identity (E1), unfamiliar face identity (E2), and face emotion (E3) is influenced by disguising the eyes with sunglasses or the mouth with masks. This was compared in people with typical recognition ability and so-called super recognizers (SR). Recognition performance was measured in a face matching task (E1-2) and a forced-choice task, i.e., tasks with minimal memory demand. The paper is overall well written and easy to read. The figures are clear and the statistical approach for the critical analysis seem sound. Below I outline major and minor questions.

Comment 1. If I understand correctly, in Experiment 1 the SR were familiar with more of the famous faces than the control groups (i.e, SR: 11/12 versus controls 9/12 = 92% vs 75%). Could this also explain why the SR were better in the actual test (Figure 1)? That is, do they score higher because they are super recognizers or do they score higher because they happened to be more familiar with the people? If the latter, it would imply that super recognizers are only perhaps super recognizers for unknown faces (or in other words, everyone is super recognizers with familiar faces)?

- **Response.** We have added text to address this comment on page 11 “This familiarity advantage may explain the difference in performance of SRs and controls”.

Comment 2. For Experiments 1-3, what does the RT data look like? Could the increased accuracy for super recognizers be explained by also longer RTs, or are they both more accurate and faster?

- **Response.** We did not record reaction time data for Experiments 1-3. We have suggested on page 23 that although group RT differences between super-recognisers and control participants have not been found in memory tasks (Belanova et al., 2018), this could be an interesting avenue for future research.

Comment 3. For the interactions in Experiment 3, the authors choose to focus on the emotion x disguise rather than the emotion x group (or both). A rationale for why one is analyzed, but not the other, could be useful. Since the graph splits the data by group, it seems strange to skip any analysis where group is involved.

- **Response.** We now report the breakdown for both the emotion x disguise and the emotion x group interactions.

Comment 4. In the introduction, it could be useful to be more specific about the novel

aspects of this study. For example, perhaps a brief summary sentence(s) in the final paragraph could state the novelty (and perhaps a rationale for why it is important)? The intro does mention novel aspects throughout, but some places it is a bit unclear what exactly is novel. For example, on the second page of the introduction (first paragraph), it is referred to research that occludes eyes with sunglasses [30], and another (preprint) study testing the effects of surgical masks [37], yet the concluding sentence of the paragraph reads “To date, no studies have compared the effect of occluding the eye region and occluding the mouth region on face matching performance”. In this particular example, is novelty the face matching task (rather than some other recognition task) or is it that both conditions are included in the same study. In either case, a rationale for why this is important could also help the readers.

- **Response.** We have added a statement to emphasise the novelty of the current work to the end of the introduction. “In sum, here we provide the first study to directly compare face matching and emotion categorisation performance for super-recognisers and typical observers for faces in no concealment, sunglasses and masks. Our study is novel in its use of real mask images (rather than computer generated facial occlusions) in the matching task. Our matching task mask stimuli are representative of the types of masks worn during COVID-19.” We have also clarified the text on page 4: “To date, no studies have directly compared the effect of each of these conditions (occluding the eye region compared against occluding the mouth region) on face matching performance”.

Comment 5. Practiced participants: When defining the ‘practiced participants’, the MS states “[Practiced participants] appeared to be particularly engaged and continued to take part in various face perception studies”? What does “particularly engaged” mean? How many experiments on average had the Ps already taken part in? Did they receive any face training during these experiments (or feedback for correct/incorrect trials)? If they only participated in standard experiments without feedback or training, then is there a rational reason to believe that they have become better at face recognition as a result of these studies (I can see that they are better at grasping the mechanics of a computerized face recognition task, but whether they have actually become better in face recognition is more difficult to apprehend)?

- **Response.** We have clarified our definition of our practiced participant group and removed the term “highly engaged” as we agree that this term is ambiguous. We describe these participants as ‘practiced’ as they have completed the CFMT+ and GFMT and at least one other face recognition test previously, and as such understand the demands of this type of research. These participants will also have been randomly invited to up to six face memory or matching projects per annum (we have also previously conducted three projects of face emotion recognition - again invitations will have been random), although due to privacy reasons we are not able to cross-reference how many they actually complete. In virtually all projects run through this database, scores are given at the end of the experiment, so participants will likely have awareness of their ability. In contrast, we describe those recruited through Prolific as ‘unpracticed’, as although it is likely they have completed a range of other research projects before, it is far less likely they will have previously completed face processing tasks.

Comment 6. In the last sentence of stimuli section for Experiment 1, it states that Ps “saw two trials in each concealment condition (...) for each trial type”. Is it correct that there were only 4 trials per condition (2 match and 2 nonmatch), or was this per face identity (thus, 4*12 trials). If the latter, did face images repeat for each celebrity? If the former, why so few trials? Relatedly, it would be good to explicitly state whether each identity was shown in each condition.

- **Response.** We have clarified this point in the manuscript on page 9. That is correct, there were only four trials in each condition for the familiar face matching task (Experiment 1). We did not repeat identities, so each of the 12 identities was only seen once by each participant (as in Experiment 2 with the unfamiliar faces). We made the decision to have a small number of trials so as not to repeat identities. The small number of identities used reflects the difficulty both in finding images of celebrities wearing face masks, and also of finding a number of celebrities who would be familiar to the majority of our participants.

Comment 7. For the correlation data (Figure 4, 5), why were not the non-practiced participants added?

- **Response.** We do not have GFMT/CFMT+ scores for non-practiced participants. These participants only completed our experimental tasks. The GFMT and CFMT+ scores for the SRs and practiced controls are taken from previous tests of these participants.

Comment 8. In the summary for Experiment 2, it is stated that one could expect no overlap between SR and practiced controls. However, given that experiments are noisy (especially true for short experiments), I would in fact have expected that some super recognizers would overlap with practiced controls just due to chance, yet that they will be different at the group level. Also that said, if the argument is kept, then it could be beneficial to report the range of scores on the CFMT. Based on the argument, there should be no overlap between the highest practiced participant and lowest SR (unless I mistook something)?

- **Response.** We have removed the comment that the reviewer has referred to as we agree it was misleading and have replaced it with “This result demonstrates noise in experimental testing [56] and speaks to the theoretical issues associated with the definition and selection of SRs [45]”. We report the range of scores on the CFMT and GFMT on page 14.

Minor Comments

Comment 9. page 4 (intro): “... task difficulty (e.g., [21] and ...”  bracket missing?

- **Response.** We have corrected this typing error.

Comment 10. The introduction refers to findings with memory tasks, but a lack of similar

findings with face matching tasks (e.g., page 2 of introduction, 2nd paragraph). It could be useful to be more specific about the memory task, if the goal is to distinguish the face matching task from the memory task. For example, what was the memory demand of the memory task (e.g., 1 second, 10 second, 30 minutes)? Note that even in a face matching task, there is a memory demand, since once can (presumably) only fixate/attend one face at the time.

- **Response.** We have added the time requirements as well as further detail of the memory tasks to the text on page 5.

Comment 11. In E1, perhaps parts of the stimuli section referring to trial structure would be more suited in the subsequent procedure section? That is, the trial structure is meaningless without knowing the paradigm, which is described in the following 'procedure' section.

- **Response.** We have moved the information regarding the assignment of identities to conditions etc. to the procedure section as suggested.

Comment 12. Experiment 1 procedure. How long were the displays on for? Timed or until response? Would be good to be more specific about the procedure, and perhaps even add a figure (e.g., as panel A for Figure 1).

- **Response.** We have added text to clarify stimuli presentation duration (until response) for Experiment 1.

Comment 13. Experiment 1, results: When referring to number of celebrities known, it could be good to use percentage or at least show the maximum possible (e.g., $M = 75\%$ (9/12)). Also, if this is a mean score, it seems strange that there are no decimals?

- **Response.** We have added the percentage with decimal places, and maximum.

Reviewer #2.

Comments to the Author(s)

In this manuscript the authors pursue the contemporary / obvious question of whether face occlusion impairs facial identity matching and emotion categorization, and whether such an impairment varies as a function of independently assessed face matching / recognition performance. Their results are as anticipated: less information available impairs performance (experiments 1 & 2), and more so if the information occluded is that which is diagnostic for the task (experiment 3). Below I detail main points, which I believe can be addressed.

Sincerely,

Meike Ramon

Comment 1. Analyses. The authors should fully exploit the fact that their observers performed all tasks, and include a more detailed, multivariate consideration of their performance patterns across all three experiments, under consideration of patterns at the individual level as they propose. Appropriate visualisations should also be provided in order

for readers to fully appreciate the depth of available information at the single-subject level. Correlational analyses should be provided throughout (not only selectively), as well as with/without inclusion of supers to address the question of whether they are driving linear relationships.

- **Response.** We have added a new section to the results ‘Associations across tasks’, which describes how performance on each task is related to performance on the others, with appropriate visualisations (page 21). We find little evidence that the relationships are driven by the inclusion of the SRs. We have included this additional analysis, both with the inclusion and exclusion of SRs, at the task level and the condition level in the supplementary analysis.

Comment 2. p.19., 2nd paragraph and Table 1: extremely cumbersome to work through; provide more straightforward approach and reporting.

- **Response.** We have restructured the results for Experiment 3 to make them easier to follow, both in the text and the table.

Comment 3. Missing methodological information

There is no information provided regarding quality assurance measures implemented to control for viewing conditions across observers, which is particularly important given data acquisition was performed online. I would like to see the required information considering restrictions implemented to prevent participation from tablets and mobile phones, as well as details regarding implemented screen calibration procedures.

- **Response.** We did not collect information about participants’ devices. Our large sample somewhat protects us from the noise associated with the relative lack of control over viewing conditions induced by online data collection. Indeed, online tests of cognitive processing can yield high-quality data, indistinguishable from that collected in the lab (e.g. see Germine et al. 2012). Given that our main question of interest is focussed on the repeated-measures variable of condition, differences in viewing conditions cannot explain the effects found.

Comment 4. No detail is provided regarding the selection of image material for experiments 1 and 2 thereby preventing reproduction.

- **Response.** We have added additional details of image selection, such as mainly front-facing images, and having no constraints on expression etc. for Experiments 1 and 2. These are in line with previous studies, now mentioned in the stimuli sections.

Comment 5. Describe how authors “ensure[d] that the intensity and validity of expressions was consistent across concealment conditions”.

- **Response.** We have made this clearer in the manuscript. The underlying facial expression was identical in the different conditions, given that we superimposed the masks onto the exact same images.

Comment 6. Diagnostic approach at odds with authors own recent practice

In a recent manuscript (Dunn et al., 2020) the GFMT and CFMT+ were described as inappropriate for online assessment, motivating the creation of an(other) online test battery. Therefore, instead of framing the manuscript as a “Super-Recognizer” study the more appropriate approach is to consider only the measure which does not suffer from a ceiling effect (ie the CFMT+), and to investigate the relationship between face recognition performance and the manipulations implemented here.

- **Response.** We have added text to clarify our reasons for using the CFMT+ and GFMT criteria for defining super-recognisers and controls in our experiment on pages 7 and 8. We followed the same practice as described in three published peer-reviewed papers using similar online recruitment methods (Correll, Ma, & Davis, 2020; Davis, Bretfelean, Belanova, & Thompson, 2020; Satchell et al., 2019). To the best of our knowledge, the CFMT+ has been employed in all published research articles to attribute super-recognition ability (albeit most published research has used lower thresholds than applied here), and following best practice, we use the GFMT as a second verifying test of face recognition ability. We acknowledge that the GFMT suffers from ceiling effects, but see it is a strength of our study that we include this second verification as research has shown that some super-recognisers (as assessed by the CFMT+ alone) score surprisingly poorly on other tests of face recognition ability, clearly questioning their status as super-recognisers.

We agree that the specific wording the reviewer refers to in the Dunn et al. pre-print concerning the use of the CFMT+, the GFMT, and the new test, the UNSW is confusing. The final paper has now been published and the edited confusing statement refers to the problem that because many researchers have employed the CFMT+ and the GFMT online, there is a risk of participants taking the tests more than once (and practice effects). However, we can assure the reviewer that those participants recruited to the current research had not previously taken the CFMT+ and GFMT before, and we have included statements to this effect in the manuscript.

Comment 7. Missing references

A number of relevant references, which should be cited are missing. These relate to the issue of familiar face processing including the aspect of partial information processing (cf. Ramon & Gobbini, 2018, *Vis Cogn*; Ramon, Busigny, Gosselin & Rossion, 2018, *Vis Cogn*), as well as those concerning the relationship across tests observed at the group vs. individual level (Stacchi et al., 2020; Fysh et al., 2020), and notably the review of super-recogniser literature by Ramon et al., 2019 (a,b; *BJP*).

- **Response.** We thank the reviewer for highlighting some additional literature and have added the following references to our revised documents (Fysh et al. 2020; Ramon et al., 2019).

Minor points

1. identification should not be used interchangeably with recognition and matching (eg first sentence in the abstract), and – in light of the reported manipulations implemented here –

refer to the tasks with greater precision (e.g. p.4, l.31: comparisons of facial *identity*; given the nature of response measurement experiment 3 is actually an emotion categorisation, rather than recognition task).

- **Response.** We have revised the manuscript accordingly.

2. p.5, 1st paragraph: the related composite face effect should be discussed, as well as its implications that facial information in the bottom of the face may have an either facilitatory as well as inhibitory effect.

- **Response.** We have included an additional description of the composite face illusion and its implications on page 3.

3. remove p.5 2nd paragraph; eye tracking in normal controls and supers not appropriate, as information sampling does not imply usage, and the prior is not assessed here.

- **Response.** We have edited this paragraph to make it clear that we are referring to suggested mechanisms that are posited by the literature in this field.

4. range of performance scores should be provided for all samples reported (cf. p.8)

- **Response.** We have made this edit to the manuscript.

5. remove reference to results reported purely in the Supplementary Material (cf. p.11, l.17f.)

- **Response.** We hesitate to remove the reference to the results in the supplementary material because we believe that this reference is important to direct readers to the supplementary material.

Additional edits

In the first version of the manuscript we used ‘super-recognisers’ and its abbreviated form SRs interchangeably. For clarity, we now refer to super-recognisers throughout.

We added reference to how our findings fit with theory on the value of the eye region in face perception to our discussion section.

Appendix B

RSOS-201168

Title: The effect of face masks and sunglasses on identity and expression recognition with super-recognisers and typical observers.

Authors: Eilidh Noyes, Josh Davis, Nikolay Petrov, Katie L.H. Gray, Kay L. Ritchie.

5/02/21

Dear Editor,

Please find uploaded the revised version of the manuscript ‘The effect of face masks and sunglasses on identity and expression recognition with super-recognisers and typical observers’ with co-authors Josh Davis, Nikolay Petrov, Katie L.H. Gray and Kay L. Ritchie. We hope that you will consider this revised paper for publication in the journal Royal Society Open Science.

We would like to thank the reviewer for their comments on the manuscript, which we have addressed in turn.

Please do not hesitate to contact me if I may be of any assistance in processing the manuscript.

Yours Sincerely,

Eilidh Noyes

Eilidh Noyes, Ph.D.

Dept. of Psychology

University of Huddersfield

Huddersfield, UK

HD1 3DH

Email: e.noyes@hud.ac.uk

Tel: 01484472770

RSOS-201168

Title: The effect of face masks and sunglasses on identity and expression recognition with super-recognisers and typical observers

Authors: Eilidh Noyes, Josh Davis, Nikolay Petrov, Katie L.H. Gray, Kay L. Ritchie.

Reviewer comments to Author:

Reviewer: 1

Comments to the Author(s)

I thank the authors for their responses and revisions. Below are my remaining comments.

- Response to comments #1 and #2: I think these limitations should at least be explicitly mentioned in the general discussion. Essentially, the enhanced performance of super recognizers in E1 could be due to they simply being more familiar with the faces (i.e., not specific to super recognizer ability). Moreover, the enhanced performance of super recognizers in general could be due to them spending more time (e.g., being more meticulous) than the other participants on each image before making their judgement.

- **Response.** We have added text to acknowledge these in the discussion. We had already included a sentence around the possibility of speed-accuracy tradeoffs (now highlighted; page 23).

- Response to comment #5: Perhaps explicitly define practiced and non-practiced in the text. E.g., (if I understand the definition of each group correctly)

o practiced: participated in previous experiments from the lab

o non-practiced: randomly recruited Ps who may or may not have done face recognition studies in the past.

Moreover, it seems unclear whether the “practiced” participants have in fact completed more face-recognition studies than the “non-practiced” participants? If I understand correctly, the practiced have done more face recognition tests offered by this specific research group, but it seems possible that the non-practiced Ps could have completed face recognition tests from other groups? If there is no way to ascertain this, then I think this limitation should also be explicitly mentioned in the discussion.

- **Response.** We had already given an extensive description at the end of the intro (page 7, now highlighted). We have added an additional definition in the methods section (page 8), and have mentioned this as a limitation to our understanding of the effect of practice in the discussion (page 23).

- Response to comment #6: Given the few trials (which seems abnormal(?)), this might also be good to point out as a potential limitation in the discussion.

- **Response.** We have now explicitly mentioned this in the discussion (page 23). We would like to note that adequate power can be achieved using few trials over many participants, or many trials over few participants (Baker et al., 2020). Given the effects appear robust, we don't think the relatively low number of trials is concerning.